# Food Safety Practices during Hajj: On-Site Inspections of Food-Serving Establishments

**DOI:** 10.3390/tropicalmed8100480

**Published:** 2023-10-23

**Authors:** Ruyuf Alnafisah, Fahad Alnasiri, Saleh Alzaharni, Ibrahim Alshikhi, Amani Alqahtani

**Affiliations:** Saudi Food and Drug Authority, Riyadh 13513, Saudi Arabia; fanasiri@sfda.gov.sa (F.A.); s.azahrani@sfda.gov.sa (S.A.); iashikhi@sfda.gov.sa (I.A.)

**Keywords:** Hajj, food-serving establishments, food safety practices, cleanliness, worker commitments, refrigerated and frozen food, food storage, professional training, professional supervision

## Abstract

The presence of crowds during Hajj increases the risk of foodborne infection. Yet, research on the practices of food handlers during Hajj is limited. This study aimed to assess compliance with food safety practices and its associated factors during Hajj 2022. An observational cross-sectional study was conducted in Mecca and Madinah before and during Hajj 2022 and involved 195 food-serving establishments (FSEs) contracted for Hajj catering. Collected data included visit time, establishment location, licensure, whether food handlers had food safety training (professional training), and whether FSEs were under supervision from a consulting office (professional supervision). The included FSEs were 168/195 (86.2%). Two-thirds of FSEs surveyed (113, 67.3%) were under professional supervision, and 91 (54.2%) hired trained food safety workers. Compliance rates varied between outcomes (72.67 ± 17.21% to 88.3 ± 18.8%). Compared to Mecca, Madinah FSEs were more adherent to cleanliness (80.5 ± 27.9% vs. 91.5 ± 19.9%, respectively, *p* = 0.006). FSEs with trained workers were more likely to comply with proper food safety practices compared to those with untrained workers: cleanliness (OR: 7.2, 95% CI [2.6–20.23], *p* < 0.001); workers’ commitment to health requirements (OR: 2.8, 95% CI [1.1–6.9], *p* = 0.025); handling of refrigerated and frozen food (OR: 5.27, 95% CI [1.83–15.20], *p* = 0.004); and food storage practices (OR: 12.5, 95% CI [2.0–12.5], *p* < 0.001). The role of professional training in increasing food safety practices compliance was highlighted. FSEs in Madinah were more compliant with food safety practices than those in Mecca. Therefore, Mecca FSEs may need stringent safety measures.

## 1. Introduction

The Hajj pilgrimage in Saudi Arabia is considered one of the largest annual mass gatherings in the world. Every year, around 2 million Muslims from over 180 countries visit Mecca in Saudi Arabia to take part in Hajj. During the days before and after the Hajj season, pilgrims tend also to visit Madinah city, where the Prophet’s Mosque is located [1]. The risk of infections, particularly foodborne infections, increases during Hajj because a large number of pilgrims from around the world congregate in a small area of high temperature for a short period of time [2,3].

Foodborne infections are a major public health concern worldwide. According to the World Health Organization (WHO), about 1 in 10 people become infected with a foodborne illness each year, leading to 420,000 deaths and 33 million disability-adjusted life years (DALYs) [4]. A systematic review identified approximately 15 outbreaks of gastrointestinal diseases that were linked to fecal–oral transmission during other large gatherings besides Hajj [5]. In the context of Hajj, foodborne disease outbreaks occur frequently, with traveler’s diarrhea being the most common [6]. The last cases of Hajj-related cholera were reported in 1989 [7]. A recent study conducted during Hajj 2019 found that 9.7% of pilgrims experienced GI symptoms and 5.1% had diarrhea [8]. In comparison, during the Hajj seasons of 2016, 2017, and 2018, approximately twice as many GI-related adverse events were reported amongst a sample of French pilgrims, with 18.6% of pilgrims experiencing at least one GI symptom, 13.8% experiencing diarrhea, and 36.4% acquiring at least one enteric pathogen [9].

Throughout the Hajj season, The Kingdom of Saudi Arabia utilizes all available resources to provide the best services to the pilgrims [10]. To control foodborne infections, the Saudi Food and Drug Authority (SFDA) inspects food imported by pilgrims through airports and seaports, as well as via travel across land, and controls food-serving establishments in Madinah and Mecca. The SFDA also inspects the processes of food transportation into Mecca and spreads awareness about the safety of food among Hajj workers. For the 2022 Hajj season, the SFDA deployed 115 specialists and inspectors to participate in the food-serving establishment control program in both Mecca and Madinah. As a result, several food-serving establishments were accessed, 146 training workshops were conducted, and more than 1,440 awareness leaflets about health requirements in five languages, including Arabic, Hindi, Urdu, Bengali, and English, were distributed for food handlers [11]. To support food safety throughout the manufacturing stages, starting with raw materials and ending with the final product, the SFDA encourages the adaptation of the Hazard Analysis and Critical Control Points (HACCP) system [12]. Additionally, food handlers working in holy cities (Mecca and Madinah) were required to adhere to certain health practices and vaccinations requirements, along with a yearly medical exam.

Although the Saudi Government has taken significant steps to prevent foodborne infections during the Hajj season, limited studies have been conducted on the adherence of food handlers to food safety requirements. The available studies highlight that between 29% and 50.2% of food handlers were afflicted with food poisoning during the Hajj seasons of 2002, 2004, and 2007 [13,14,15]. Food handlers are not, however, the subject of any recent studies. To the best of our knowledge, no investigation has been conducted into the adherence of food-serving establishments and their workers to food safety practices during the Hajj. Therefore, this study aimed to assess the compliance of food-serving establishments with food safety practices and to determine the factors associated with their compliance during the 2022 Hajj season.

## 2. Materials and Methods

### 2.1. Study Design

An on-site, observational, cross-sectional study was conducted in food-serving establishments located in Mecca and Madinah before and during the Hajj pilgrimage. As the peak days of the Hajj occur each year between the 8th and 12th of Dhul-Hijjah (as per the lunar calendar), the corresponding Gregorian dates for 2022 were between the 2nd and 12th of July. In accordance, this study was conducted from May to June 2022 in Madinah and from June to July 2022 in Mecca.

### 2.2. Targeted Sample and Recruitment

The study targeted all contracted food-serving establishments that were registered for food catering during the 2022 Hajj season (195 establishments). These establishments were categorized into four distinct groups based on their municipal licenses: catering contractors; kitchens for banquets and parties; hotels; and restaurants and food stores. This categorization was independently conducted by the respective municipal authorities based on each establishment’s size and activity. The included establishments were visited by ten inspectors (seven in Mecca and three in Madinah), who collected data through on-site inspections. The inspectors were well qualified. Each inspector had at least a bachelor’s degree in a discipline closely aligned with nutrition and food sciences, showcasing a strong academic foundation relevant to the subject matter of this study. Importantly, these inspectors had collectively accrued a minimum of two years of practical experience in the field of food inspection. Moreover, this study was part of the inspection campaign run by the SFDA; therefore, the day of the visit was not disclosed to the establishments and their workers. Following the completion of the data collection process, the data were then reviewed by a team of three specialists to guarantee their validity.

### 2.3. Collection of Data

The collected data included general information related to the time of visit and establishment location. The survey also directly asked whether food handlers received professional training courses on food safety either online or in person (professional training) and if they were under the supervision of a consulting office (professional supervision). Consulting offices were generally responsible for conducting food-safety-related activities, such as supervising activities along the food supply chain and offering food management systems, such as HACCP, Good Manufacturing Practices (GMPs), and Good Hygienic Practices (GHPs).

To assess food safety practices, the survey included a checklist of 41 items that was developed according to the SFDA technical regulation “Hygienic Regulation for Food-serving Establishments and their Workers” [SFDA.FD 21/2018] [16], which was adopted based on the international standards developed by the FAO/WHO Codex Alimentarius [17].

The checklist was then divided into four major outcomes (Table 1). The first outcome contained nine items related to “Cleanliness”, in which the condition of the equipment and utensils used and the areas of food handling, washing, disinfecting, and drying processes were assessed. The second outcome included eight items that were used to assess the “Commitment of workers to health requirements”. These health requirements focused on how workers act and dress, how well their establishments are maintained, and how they deal with potential issues, such as disease outbreaks and open wounds (if present). The third outcome included nine items regarding “Handling refrigerated and frozen food”. Such items evaluated the conditions of freezing units and refrigerators and their temperature gauges, dealing with refrigerated and frozen foods, and the sources of meat sacrifices, as the slaughter of animals is part of Hajj rituals. The last outcome included 15 items related to “Food storage practices”, which mainly focused on the receiving and storing circumstances, processes, and labeling practices.

Depending on whether the observed item conformed to the technical regulation or not, all items were answered with a yes or no response (score 1/0). The scores for each outcome were then added together to create a final score, which was then transformed to a percentage to ensure standardization across outcomes. The final scores were divided into two categories based on the mean value. Values below the average were classified as “low compliance”, while values equal to or above the mean were labeled as “high compliance”.

After this survey was developed, an expert panel revised and approved the survey before it was administered. The expert panel assessed the survey’s content, questions, and checklist items to ensure that they were relevant and aligned with this study’s objectives.

### 2.4. Outcomes Internal Reliability

To assess the internal consistency of the survey, Cronbach’s alpha was used for each outcome. This statistical tool helps to verify that the items under different constructs deliver consistent results. The interpretation of Cronbach’s alpha was based on the rules of thumb provided by George and Mallery (2003): α > 0.9 (excellent), >0.8 (good), >0.7 (acceptable), >0.6 (questionable), >0.5 (poor), and <0.5 (unacceptable) [18].

The results of this study show that the interitem correlations varied from 0.66 to 0.88. Items under the outcomes “Cleanliness” and “Food Storage Practices” exhibited good internal consistency, while the internal consistency of items under the outcomes “Commitment of Workers to Health Requirements” and “Handling Refrigerated and Frozen Food” was acceptable (Table 2).

### 2.5. Statistical Analysis

Data were analyzed using SPSS statistical software version 29.0. The normality of all quantitative variables was tested before the analysis was performed. Mean and standard deviations were used to quantify the quantitative variables, while an independent *t*-test was used (when appropriate) to compare their mean values across the categories of study variables. For the categorical variables, frequencies and percentages were described, and a Chi-squared test was used to compare the difference in proportions. To identify the factors associated with the compliance with proper food safety practices, a binary logistic regression was performed using establishment location, professional supervision, and training as independent factors and the outcomes “Cleanliness”, “Commitment of workers to health requirements”, “Handling refrigerated and frozen food”, and “Food storage practices” as dependent variables. To quantify the associations between independent variables and study outcomes, odds ratios (ORs), 95% confidence intervals (CIs), and *p*-values are presented.

## 3. Results

### 3.1. General Characteristic of Food-Serving Establishments

Most of the targeted food-serving establishments during Hajj were included in the analysis (Table 3). Of these, about three-quarters were located in Mecca. The majority of establishments were catering contractors. Compared to Madinah, food-serving establishments in Mecca were significantly higher in providing professional supervision and hiring workers with professional training.

### 3.2. Compliance with Food Safety Practices

The average compliance rates for food safety practices ranged between 83.3 ± 26.5% and 87.6 ± 17.4% (Table 4). The rates of compliance with cleanliness practices between establishments in Mecca and Madinah were significantly different.

### 3.3. Factors Associated with Compliance with Food Safety Practices

Madinah reported more compliance to proper food safety practices related to cleanliness, commitment of workers to health requirements, and handling refrigerated and frozen food. Establishments that hired workers with professional training were more likely to comply to proper food safety practices related to all outcomes (Table 5).

## 4. Discussion

To the best of our knowledge, this is the first study to evaluate the compliance of food-serving establishments and their workers with food safety practices during the Hajj season. Overall, the findings indicated that food-serving establishments generally complied with proper food safety practices, with adherence rates ranging from 72.7% to 88.3%.

This study found that cleanliness practices were generally well maintained (83.27%). These findings are consistent with a prior investigation carried out in the United Kingdom (UK) during the 2012 Olympics, where 66% of the establishments surveyed were discovered to have appropriate hygiene food safety practices. However, approximately 8% of the collected food samples from the UK study were deemed to have poor microbiological quality [19]. This suggests that even with acceptable levels of food safety practices, catering at large gatherings can cause a greater risk of food poisoning compared to that in other settings. In fact, there is evidence of between 9 and over 55,000 foodborne cases per 100,000 attendees at large gatherings besides Hajj [5]. However, the number of reported cases during Hajj seasons earlier in the decade only reached between 44 and 132 during each Hajj season [20]. In light of managing millions of pilgrims during Hajj seasons, it is clear that Saudi Arabia has gained significant expertise in the provision of healthcare at large gatherings. This achievement is attributed to various sectors involved in the annual activities in preparations for Hajj, including risk assessment, medical infrastructure and information technology utilization, and surveillance systems [21], along with the SFDA’s role in inspecting and controlling food and spreading awareness about food safety.

Other factors, beyond large gatherings, can influence the quality and safety of prepared food, such as monitoring the temperature of food [19]. In the present study, compliance with food safety practices pertaining to handling refrigerated and frozen food, which included measuring the temperatures of all kinds of meat (when the product was received), was deemed satisfactory (72.68%). The WHO emphasizes that storing food without considering temperature requirements poses a significant risk factor for foodborne diseases [22]. It has also been documented that keeping food in temperatures within the danger zone range (4.44 °C to 66 °C) for approximately 20 min leads to bacterial growth, thereby increasing the risk of foodborne illnesses [23]. Thus, temperature control stands as one of the most effective methods for ensuring food safety. Unfortunately, poor temperature control remains a major barrier to implementing HACCP and food safety regulations in food-serving establishments [24]. It is reasonable to hypothesize that inadequate temperature control is more likely to occur in the hands of less knowledgeable food handlers. Previous studies revealed that the majority of food handlers are unaware of the fundamental temperature control standards required to prevent food contamination [25,26]. Therefore, it is recommended that targeted measures be implemented to enhance food handlers’ awareness of the food danger zone temperatures. In the context of the two holy cities, the majority of establishments reported hiring workers who had received prior training in food safety practices (72%). This fact might serve as evidence for implementing efforts that aim to improve food handlers’ understanding of the danger zone temperature range in different food categories.

Notably, the result of this study revealed that professional training courses provided to food handlers were significantly associated with an increase in compliance with food safety practices. Earlier research showed that food establishments with managers who underwent food hygiene training were more likely to produce food of sufficient microbiological quality [27]. This may suggest that mandatory certification for both managers and food handlers can help to improve food safety and quality. It has been speculated that food safety practices can be highly influenced by employees’ attitudes, beliefs, and motivation [28]. Therefore, training programs can be improved by focusing on certain aspects that have an impact on food handlers’ behavior instead of only attempting to increase their knowledge [28].

This study also found that establishments in Madinah were more likely to adhere to proper food safety practices than those located in Mecca. There are several possible explanations for this result. First, visiting the Prophet’s Mosque is not part of Hajj rituals, and thus, pilgrims are not required to visit Madinah. In contrast, they must perform all Hajj rituals in Mecca on a specific date based on the lunar calendar, which leads to more people congregating in Mecca during a short period of time. Previous studies highlighted that high customer volume is one of the most significant factors affecting adherence to food safety standards [29]. Another possible explanation is that all the hotels included in this study are located in Madinah and account for around half of the city’s food-serving establishments (45.2%). Evidence from earlier research suggests that larger hotels are more likely to follow the HACCP system and other food safety strategies [30]. Therefore, the present study highlights the importance of food-serving establishments adopting the HACCP system to raise the standards of food handling practices.

There are some limitations to this study. First, this is a cross-sectional study; therefore, no follow-up study was conducted to determine whether food safety practices changed over the course of the Hajj pilgrimage (before vs. during Hajj pilgrimage). Secondly, the comparison of current findings with those of previous studies was difficult due to the scarcity of earlier research. Finally, several factors, including the size of establishments and the sociodemographic profiles, knowledge, and attitude of food handlers, were not considered during the data collection phase. Despite these limitations, this study provides a valuable contribution to the field of food safety. It is the first study that the SFDA has carried out as part of its yearly inspections of food-serving establishments during the Hajj season. Furthermore, the current study is one of the few to evaluate food safety practices in the context of a mass gathering. Additionally, data were gathered for this study through the observation of experienced inspectors rather than relying solely on self-reported surveys. This approach allowed the inspectors to compare the practices of establishments against a recently developed technical regulation (SFDA.FD 21/2018) [16].

## 5. Conclusions

The current study provides valuable insights into food safety practices during Hajj, and its results can be used to improve food safety and prevent foodborne illnesses among pilgrims. The disparity in compliance with food safety practices between establishments in Madinah and Mecca highlights the impact of crowdedness on food safety measures. Future studies are needed to investigate the microbiological quality of randomly selected food samples. Further research may also be required to assess the safety of frozen and refrigerated food items, which involves assessing the adequacy of refrigeration and storage facilities.

## Figures and Tables

**Table 1 tropicalmed-08-00480-t001:** Checklist content (outcomes/items).

Cleanliness Practices
The special area for washing and disinfecting equipment and utensils is in good condition and divided into three parts (washing, rinsing, disinfection);
Food contact surfaces, cutting surfaces, and refrigerator handles are in good condition, do not require maintenance, and are cleaned and disinfected regularly;
The utensils used are not damaged or worn out;
The utensils used are made of stainless materials and do not cause any pollution;
Equipment and utensils are cleaned immediately after use;
Equipment and utensils are cleaned in suitable sinks away from production areas;
Equipment and utensils are dried at room temperature;
Equipment and utensils are dried in a clean place isolated from contaminants and dust;
Equipment and utensils are stored in appropriate places away from dust and pollutants.
**Commitment of workers to health requirements section**
Workers wear head coverings, hand gloves, and masks;
Food is eaten outside the kitchen;
Smoking or other unacceptable practices are not allowed in the kitchen;
There are a sufficient number of toilets in clean condition and completely isolated from food handling areas;
The private area for workers to rest and eat their meals is in good condition;
Any worker with diseases and/or infected wounds is prevented from working and handling food;
Workers’ wounds are covered properly;
Workers’ diseases are documented with results, certificates, and reports.
Handling refrigerated and frozen food section
The establishment has a freezing unit;
Refrigerators and freezers contain a temperature record;
There is no accumulation of food inside the coolers and freezers;
Food products with strong odors are stored separately;
Meat sacrifices are all from trusted sources;
The temperature of all types of meat is taken before meat is received;
All refrigerated and frozen foods are valid and have not expired;
Frozen foods are thawed in refrigerators through the correct scientific method;
Thermometers used for food, refrigerators, and freezers are calibrated, cleaned, and disinfected before each use, and they are calibrated annually.
Food storage practices section
The receiving area is clean;
The receiving area is free from insects and rodents;
The receiving area is sealed;
Food items (dry materials) are received at room temperature;
Food items (refrigerated items) are received at a temperature between 0 and 4 °C;
Food items (frozen items) are received at a temperature below −18 °C;
Foods are stored in the appropriate place with sufficient space immediately upon their receiving;
Packaged and canned foods are free from holes, rupture, rust, swelling, and dust;
Places for receiving or storing raw materials are separated from places for preparing or packing the final product;
All shelves and pallets in the storage area are made of stainless materials;
Food items are stored on shelves and pallets separately in a way that allows air circulation between foods in refrigerators/freezers;
Foodstuffs are supplied from trusted suppliers;
Food items are covered in refrigerators/freezers tightly;
All stored food items contain an ingredient and expiration label;
Dry foods, refrigerated foods, and frozen foods are stored at 25 °C, 4 °C, and −18 °C temperatures, respectively.

**Table 2 tropicalmed-08-00480-t002:** The internal consistency of items in each section.

Section	Number of Items	Reliability (Cronbach’s Alpha)	Internal Consistency
Cleanliness	9	0.88	Good
Commitment of workers to health requirements	8	0.66	Acceptable
Handling refrigerated and frozen food	9	0.68	Acceptable
Food storage practices	15	0.88	Good

**Table 3 tropicalmed-08-00480-t003:** General characteristics of food-serving establishments.

Variable		All (168, 100%)	Mecca (126, 75.0%)	Madinah (42, 25.0%)	*p*-Value ^1^
Establishment category—n (%)	Catering contractors	136 (80.9%)	11 9 (94.4%)	17 (40.5%)	<0.001
	Kitchens for banquets and parties	6 (3.6%)	1 (0.8%)	5 (11.9%)	
	Hotels	19 (11.3%)	0 (0%)	19 (45.2%)	
	Restaurants and food stores	7 (4.2%)	6 (4.8%)	1 (2.4%)	
Professional supervision—n (%)		113 (67.3%)	91 (72.2%)	22 (52.4%)	0.018
Professional training—n (%)		121 (72.0%)	101 (80.2%)	20 (47.6%)	<0.001

^1^ *p*-value was obtained using a Chi-squared test.

**Table 4 tropicalmed-08-00480-t004:** Food safety practices.

Outcomes	All (168)—Mean ± SD	Mecca (126)—Mean ± SD	Madinah (42)—Mean ± SD	*p*-Value ^1^
Cleanliness	83.3 ± 26.5	80.5 ± 27.9	91.5 ± 19.9	0.006
Commitment of workers to health requirements	87.6 ± 17.4	87.2 ± 16.4	88.7 ± 20.3	0.542
Handling refrigerated and frozen food	72.7 ± 17.2	70.9 ± 17.0	78.1 ± 16.9	0.596
Food storage practices	88.3 ± 18.8	87.8 ± 19.6	89.7 ± 16.5	0.125

^1^ *p*-Value was obtained using a Mann–Whitney U test.

**Table 5 tropicalmed-08-00480-t005:** Factors associated with the compliance of food safety practices.

Variable	Cleanliness (122, 72.6%) ^2^		Commitment of Workers to Health Requirements (84, 50%) ^2^		Handling Refrigerated and Frozen Food (82, 48.8%) ^2^		Food Storage Practices (111, 66.1%) ^2^	
	OR ^1^ [95% CI]	*p*-Value	OR ^1^ [95% CI]	*p*-Value	OR ^1^ [95% CI]	*p*-Value	OR ^1^ [95% CI]	*p*-Value
**Establishment location ^3^**	4.9 [1.7–14.2]	** 0.003 **	2.5 [1.1–5.6]	** 0.024 **	7.1 [2.6–18.9]	** <0.001 **	1.0 [0.4–2.2]	0.963
**Professional training ^4^**	7.2 [2.6–20.3]	** <0.001 **	2.8 [1.1–6.9]	** 0.025 **	5.3 [1.8–15.2]	** 0.004 **	12.5 [2.0–12.5]	** <0.001 **
**Professional supervision ^5^**	0.8 [0.3–2.0]	0.583	1.1 [0.5–2.5]	0.744	1.3 [0.56–3.0]	0.556	0.9 [0.4–2.1]	0.777

^1^ Odds ratio (OR) was obtained from a binary logistic regression; low compliance = 0; high compliance = 1. ^2^ High compliance (n, %). ^3^ Reference: Mecca. ^4^ Reference: No professional training. ^5^ Reference: No professional supervision.

## Data Availability

Data available on request.

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
