# Peer review of "Food Safety Practices during Hajj: On-Site Inspections of Food-Serving Establishments"

_tropicalmed, 2023, doi:10.3390/tropicalmed8100480_

Round 1

Reviewer 1 Report

The comments are included in the PDF attached.

Reviewer 2 Report

Food Safety Practices during the Hajj: on-site Inspections of Food-Serving Establishments

To assess food safety practices, the survey included a checklist of 41 items that was developed according to SFDA technical regulation. The results presented in table 1 do not allow to know if important factors that may possess an undeviating impact on the hygienic condition of the restaurant were considered. The study assessed the overall housing condition of the restaurant by observing its location, however, physical evaluation as dining room space, ventilation, door system, lighting, and room temperature were not mentioned, as well as all critical sections or areas in the premise such as the processing and cooking area, dining and serving area.

There are three most significant contributors to food-borne illnesses in restaurants include time-temperature abuse, personal hygiene, and cross contamination (US Food and Drug Administration, 2009).

It is not specify if the following factors were considered as well:

-       proper hot meals holding

-       liquid and solid waste disposal system properly maintained

-       use uncovered and overflowded bins located beside food storage place

-       the raw materials and finished products, processing plant, manufacturing flow and the handling aspect during production, packaging materials, quality control and assurance practice

-       storage practices: food containers labelled

-       staff hygiene: condition of fingernails, and maintaining proper dress code like wearing apron and suitable attire while handling food

-       Pest control system

Reviewer 3 Report

This manuscript can be published after minor changes.

Author Response

Thank you very much for taking the time to review this manuscript. 

Reviewer 4 Report

The article by Alnafisah et al. presents the results of on-site inspections of food-serving establishments during the Hajj in 2022. Though the attitude of food handlers and the surrounding hygiene level of the catering establishments are key factors for food safety during massive gatherings such as Hajj, the manuscript according to my opinion exhibits critical methodological shortcomings that render it unsuitable for consideration to be published in its current form. Overall, the structure of the study lacks clarity and scientific accuracy as currently presented in the manuscript. For example:

·      -   necessary details for the qualifications of the 10 inspectors (in order to justify their competence and, therefore, the objectivity of their evaluations) are not provided.

·     -    the criteria by which the establishments were grouped into 4 categories are vague and there is no justifiable correlation of this categorisation with the single checklist used.

·     -    the criteria/methodology by which the 41 items of the checklist were chosen and grouped based on the SDA technical regulation are not presented in the manuscript.

·     -    the tool (checklist) that was used for the collection of data cannot be reproduced by other studies due to the lack in detail and clarity of its actual structure and content (e.g. what does the activity of the license mean?, how many questions were relevant to whether the food handlers received professional training in the past, did quality managements systems include or not food safety management systems?, which quality assurance systems do the authors refer to?,  etc).

·     -    the same numerical results are duplicated both in the main text and the tables and the phrase ‘More details in Table …’ is inappropriate for a scientific article.

The English language used is incomprehensible in numerous points (e.g. what do ‘food temperature measures’ mean?)  and  the terms ‘measurements’, ‘survey’, and ‘outcomes’ are used inappropriately since they don’t correspond to their actual meaning.

Round 2

Reviewer 2 Report

There are no comments.

Author Response

(The authors gave the same response as above.)

Reviewer 4 Report

The revised article by Alnafisah et al. is a better version of the initially submitted manuscript. However, several spelling and content revisions are still necessary to be eligible for publication according to the following comments:

Line 99: Delete “Measurements” and rename the title “Collection of data”. .

Lines 105-106: The term “food quality assurance” is misleading in the context of the study and should be deleted. It is generally very difficult to assure the quality of food since food safety is always a part of food quality but every single item of food produced cannot be analysed and verified for its safety (there is always a possibility that food safety hazards are present above the allowed limits in foodstuffs). Instead, only food safety management systems can be applied and certified according to international (e.g. ISO 22000, FSSC 22000, BRC, IFS) or private standards. The combination of GHP/GMP+HACCP is the simplest FSMS that can be applied according to Codex Alimentarius. Therefore, GHP/GMP+HACCP are usually the minimum legislative requirements in most countries. Based on the above, the authors should clarify here if the establishments investigated were just following the law (application of GHP/GMP+HACCP) and if they were additionally certified according to an international FSMS (e.g. ISO 22000:2018) or not.  

Lines 110-112: The yellow highlighted sentence is written in poor English and reference 17 (EU Regulation 852/2004) is not relevant to the content of this sentence. A correct reference should be added. Moreover, the international guidelines for food safety belong to FAO/WHO Codex Alimentarius, not just FAO as indicated in the manuscript.

Page 4: All the checklist items should be written in proper English and following the same style (the hygienic Regulation naturally contains sentences with the term “should” but these requirements should be correctly transformed to checklist items by not including the term ‘should’). For example, “Packaged and canned food should be free from holes, rupture, rust, swelling and dust;”

Likewise, the following items need appropriate rephrasing according to the example: “Prevent any worker with diseases and/or infected wounds from working and handling food” must be rephrased to “Any worker with diseases and/or infected wounds is prevented from working and handling food”:

·         Cover workers' wounds properly;

·         Take the temperature of all types of meat before receiving it;

·         All refrigerated and frozen foods are valid and have not expired;

·         Receive food items (dry materials) at room temperature;

·         Upon receiving it, storing food in the appropriate place with sufficient space immediately;

·         Packaged and canned food should be free from holes, rupture, rust, swelling and dust;

·         Store food items on shelves and pallets separately in a way that allows air circulation between foods in refrigerators/freezers;

·         Supplying foodstuffs from trusted suppliers;

·         Cover food items in refrigerators/freezers tightly

The authors have responded that “Adjustments are made as per the comments in the results part”, but the same numerical results are still duplicated both in the main text and the tables. For example, in Line 168 “Food-serving establishments in Mecca (91, 72.2%, p=0.018)” and the same numbers again appear in Table 3.

Though the authors responded that "The manuscript has been revised by professional native English-speaking experts and adjustments have been made accordingly”, there are still several sentences written in poor English in the revised manuscript (e.g., Lines 110-112) as indicated in the main comments. 

Author Response

For research article

Response to Reviewer X Comments

1. Summary

Thank you very much for taking the time to review this manuscript. Please find the detailed responses below and the corresponding revisions/corrections in track changes in the re-submitted files.

2. Questions for General Evaluation

Reviewer’s Evaluation

Response and Revisions

Does the introduction provide sufficient background and include all relevant references?

Yes/Can be improved/Must be improved/Not applicable

Are all the cited references relevant to the research?

Yes/Can be improved/Must be improved/Not applicable

The pointed reference was adjusted

Is the research design appropriate?

Yes/Can be improved/Must be improved/Not applicable

All comments were taken into consideration

Are the methods adequately described?

Yes/Can be improved/Must be improved/Not applicable

All comments were taken into consideration

Are the results clearly presented?

Yes/Can be improved/Must be improved/Not applicable

All comments were taken into consideration

Are the conclusions supported by the results?

Yes/Can be improved/Must be improved/Not applicable

3. Point-by-point response to Comments and Suggestions for Authors

Comments 1: Line 99: Delete “Measurements” and rename the title “Collection of data”.

Response 1: Thank you for pointing this out. We agree with this comment. Therefore, the title was changed

Comments 2: [Lines 105-106: The term “food quality assurance” is misleading in the context of the study and should be deleted. It is generally very difficult to assure the quality of food since food safety is always a part of food quality but every single item of food produced cannot be analysed and verified for its safety (there is always a possibility that food safety hazards are present above the allowed limits in foodstuffs). Instead, only food safety management systems can be applied and certified according to international (e.g. ISO 22000, FSSC 22000, BRC, IFS) or private standards. The combination of GHP/GMP+HACCP is the simplest FSMS that can be applied according to Codex Alimentarius. Therefore, GHP/GMP+HACCP are usually the minimum legislative requirements in most countries. Based on the above, the authors should clarify here if the establishments investigated were just following the law (application of GHP/GMP+HACCP) and if they were additionally certified according to an international FSMS (e.g. ISO 22000:2018) or not.]

Response 2: Agree. We have, accordingly, deleted the term “food quality assurance” 

Comments 3: [Lines 110-112: The yellow highlighted sentence is written in poor English and reference 17 (EU Regulation 852/2004) is not relevant to the content of this sentence. A correct reference should be added. Moreover, the international guidelines for food safety belong to FAO/WHO Codex Alimentarius, not just FAO as indicated in the manuscript..]

Response 3: Agree. We have, accordingly, adjusted the sentence as well as the reference

Reference:

FAO. Regulation (EC) No. 852/2004 of the European Parliament and of the Council on the hygiene of foodstuffs. Off J Eur Union [Internet]. 2004;3:54–54. Available from: https://www.fao.org/faolex/results/details/en/c/LEX-FAOC063426/#:~:text=This Regulation sets out provisions,exported and re-exported food.

Comments 4: [Page 4: All the checklist items should be written in proper English and following the same style (the hygienic Regulation naturally contains sentences with the term “should” but these requirements should be correctly transformed to checklist items by not including the term ‘should’). For example, “Packaged and canned food should be free from holes, rupture, rust, swelling and dust;”

Likewise, the following items need appropriate rephrasing according to the example: “Prevent any worker with diseases and/or infected wounds from working and handling food” must be rephrased to “Any worker with diseases and/or infected wounds is prevented from working and handling food”:

·         Cover workers' wounds properly;

·         Take the temperature of all types of meat before receiving it;

·         All refrigerated and frozen foods are valid and have not expired;

·         Receive food items (dry materials) at room temperature;

·         Upon receiving it, storing food in the appropriate place with sufficient space immediately;

·         Packaged and canned food should be free from holes, rupture, rust, swelling and dust;

·         Store food items on shelves and pallets separately in a way that allows air circulation between foods in refrigerators/freezers;

·         Supplying foodstuffs from trusted suppliers;

·         Cover food items in refrigerators/freezers tightly.]

Response 4: Thank you for pointing this out. We agree with this comment. Therefore, adjustments were made

Comments 5: [The authors have responded that “Adjustments are made as per the comments in the results part”, but the same numerical results are still duplicated both in the main text and the tables. For example, in Line 168 “Food-serving establishments in Mecca (91, 72.2%, p=0.018)” and the same numbers again appear in Table 3..]

Response 5: Thank you for making it clearer. Duplicated numbers are now removed. 

4. Response to Comments on the Quality of English Language

Point 1: Though the authors responded that "The manuscript has been revised by professional native English-speaking experts and adjustments have been made accordingly”, there are still several sentences written in poor English in the revised manuscript (e.g., Lines 110-112) as indicated in the main comments.

Response 1: The manuscript has undergone English language editing by MDPI. The text has been checked for correct use of grammar and common technical terms, and edited to a level suitable for reporting research in a scholarly journal (attached is the certificate). 
